# Superior Cerebellar Atrophy: An Imaging Clue to Diagnose *ITPR1*-Related Disorders

**DOI:** 10.3390/ijms23126723

**Published:** 2022-06-16

**Authors:** Romina Romaniello, Ludovica Pasca, Elena Panzeri, Fulvio D’Abrusco, Lorena Travaglini, Valentina Serpieri, Sabrina Signorini, Chiara Aiello, Enrico Bertini, Maria Teresa Bassi, Enza Maria Valente, Ginevra Zanni, Renato Borgatti, Filippo Arrigoni

**Affiliations:** 1Neuropsychiatry and Neurorehabilitation Unit, Scientific Institute, IRCCS Eugenio Medea, Bosisio Parini, 23842 Lecco, Italy; 2Department of Brain and Behavioral Sciences, University of Pavia, 27100 Pavia, Italy; ludovica.pasca01@universitadipavia.it (L.P.); renato.borgatti@mondino.it (R.B.); 3Child Neurology and Psychiatry Unit, IRCCS Mondino Foundation, 27100 Pavia, Italy; sabrina.signorini@mondino.it; 4Laboratory of Molecular Biology, Scientific Institute, IRCCS Eugenio Medea, Bosisio Parini, 23842 Lecco, Italy; elena.panzeri@lanostrafamiglia.it (E.P.); mariateresa.bassi@lanostrafamiglia.it (M.T.B.); 5Department of Molecular Medicine, University of Pavia, 27100 Pavia, Italy; fulvio.dabrusco01@universitadipavia.it (F.D.); enzamaria.valente@unipv.it (E.M.V.); 6Unit of Neuromuscular and Neurodegenerative Disorders, Department of Neurosciences, Bambino Gesù Children’s Hospital, IRCCS, 00146 Rome, Italy; lorena.travaglini@opbg.net (L.T.); chiara.aiello@opbg.net (C.A.); enricosilvio.bertini@opbg.net (E.B.); ginevra.zanni@opbg.net (G.Z.); 7Neurogenetics Research Center, IRCCS Mondino Foundation, 27100 Pavia, Italy; valentina.serpieri01@universitadipavia.it; 8Radiology and Neuroradiology Department, Children’s Hospital V. Buzzi, 20154 Milan, Italy; filippo.arrigoni@asst-fbf-sacco.it

**Keywords:** cerebellar atrophy, MRI, ataxia

## Abstract

The inositol 1,4,5-triphosphate receptor type 1 (*ITPR1*) gene encodes an InsP_3_-gated calcium channel that modulates intracellular Ca^2+^ release and is particularly expressed in cerebellar Purkinje cells. Pathogenic variants in the *ITPR1* gene are associated with different types of autosomal dominant spinocerebellar ataxia: SCA15 (adult onset), SCA29 (early-onset), and Gillespie syndrome. Cerebellar atrophy/hypoplasia is invariably detected, but a recognizable neuroradiological pattern has not been identified yet. With the aim of describing *ITPR1*-related neuroimaging findings, the brain MRI of 14 patients with *ITPR1* variants (11 SCA29, 1 SCA15, and 2 Gillespie) were reviewed by expert neuroradiologists. To further evaluate the role of superior vermian and hemispheric cerebellar atrophy as a clue for the diagnosis of *ITPR1*-related conditions, the *ITPR1* gene was sequenced in 5 patients with similar MRI pattern, detecting pathogenic variants in 4 of them. Considering the whole cohort, a distinctive neuroradiological pattern consisting in superior vermian and hemispheric cerebellar atrophy was identified in 83% patients with causative *ITPR1* variants, suggesting this MRI finding could represent a hallmark for *ITPR1*-related disorders.

## 1. Introduction

The inositol 1,4,5-triphosphate receptor type 1 (*ITPR1*) gene encodes a ligand-gated calcium channel localized in the endoplasmic reticulum membrane, which modulates intracellular Ca^2+^ release, representing a signaling hub for the cell [1,2]. *ITPR1* is widely expressed in the brain, with highest expression in cerebellar Purkinje cells. Heterozygous pathogenic variants of *ITPR1* have been associated with a broad clinical spectrum, ranging from adult-onset Spinocerebellar Ataxia type 15 (SCA15) to early-onset Spinocerebellar Ataxia type 29 (SCA29) and Gillespie syndrome, while homozygous variants have been described only in the early onset form [3]. SCA29 is characterized by onset in infancy or childhood of ataxia and other cerebellar signs such as hypotonia and oculomotor abnormalities, frequently associated with delayed motor development and cognitive impairment [1]. Gillespie syndrome is characterized by a more severe phenotype featuring the cerebellar syndrome, intellectual disability, and aniridia [3]. Both familial and sporadic cases with early-onset cerebellar ataxia associated with *ITPR1* gene mutations have been reported to date [4]. According to available studies, brain magnetic resonance imaging (MRI) of *ITPR1*-related disorders is characterized by cerebellar atrophy as the main finding. Extra-cerebellar findings have been rarely reported, involving both supratentorial regions and pons [4,5,6,7,8]. Most of the studies describe a pattern of cerebellar atrophy which is more severe in the vermis than in the cerebellar hemispheres, while only one patient with a pattern consistent with Ponto Cerebellar Hypoplasia (i.e., “dragonfly” cerebellum) has been reported so far [8]. In some cases, a progression of cerebellar atrophy over time is observed [2,4,6,7,9]. More recently, the pattern of atrophy was noted to be more severe in the superior part of the hemispheres (and vermis) than in the inferior part [4,6,9]. According to these observations, we reviewed a cohort of *ITPR1*-mutated patients to better define the characteristics of cerebellar alterations. In addition, genetic analysis of the *ITPR1* gene was performed on patients with superior cerebellar atrophy at MRI to establish the value of this sign as a predictive clue for the diagnosis of *ITPR1* gene-related disorders.

## 2. Materials and Methods

Three different Italian centers participated in this retrospective study: Scientific Institute E. Medea, Bosisio Parini (LC), Bambino Gesù Children’s Hospital, and Romeand Neurological Scientific Institute C. Mondino, Pavia. Two different groups of subjects were recruited. Group A included patients already diagnosed with a pathogenetic *ITPR1* gene variant through a clinical diagnostic test, while Group B included subjects without a genetic diagnosis, who presented a brain MRI pattern of isolated superior hemispheric and vermian cerebellar atrophy, identified through an extensive review of our neuroradiological databases. All the available images of Group A patients were collected and reviewed independently by two experts in pediatric neuroradiology to assess the cerebellar and cerebral findings. In particular, sagittal and coronal slices were used to evaluate the presence, distribution, and severity of cerebellar atrophy in the upper and lower cerebellum. Signal alterations of cerebellar cortex and white matter as well as supratentorial findings were also investigated. When multiple studies were available, the progression of atrophy was recorded. Patients of Group B were first identified by querying the neuroimaging databases, selecting those patients affected by predominantly or exclusively superior cerebellar atrophy with no history of acquired conditions (e.g., infective, ischemic etc.). Images were then reviewed to confirm the pattern of cerebellar atrophy predominantly involving the superior part of the hemisphere and vermis and selected subjects underwent genetic analysis of the *ITPR1* gene. Only selected patients for whom DNA material was available were finally included in Group B.

For genetic studies, genomic DNA was extracted from peripheral blood of the patients and their parents. Next Generation Sequencing (NGS) analysis was performed either on targeted panels of genes causative of various forms of cerebellar ataxias (16 patients), clinical exome (1 patient), or whole exome (2 patients). The targeted gene panels were sequenced using either Nextera (Illumina, San Diego, CA, USA) or SureSelect (Agilent Technologies, Santa Clara, CA, USA) enrichment protocols and run on MiSeq or NextSeq sequencing platforms (Illumina, San Diego, CA, USA), with an expected coverage of >99% of targeted genomic regions. For clinical exome and whole exome, DNA libraries were amplified using the SureSelectXT Focused Exome (Agilent Technologies) and Twist Human Core Kit (Twist Bioscience, South San Francisco, CA, USA), respectively, and sequenced on a NextSeq platform (Illumina). Bioinformatic analysis was carried out by aligning sequences to the human reference genome (GRCh37) using Bowtie2 or BWA v0.7.5. ANNOVAR and GATK Unified Genotyper were used to call variants, which were annotated through the eVANT v1.3 software (enGenome, Pavia, Italy). Subsequent filtering steps allowed to exclude intronic variants, synonymous variants not affecting splicing, and variants with frequency >  1% in human variation databases. We used several in silico tools to predict pathogenicity of identified variants, including Deleterious Annotation of genetic variants using Neural Networks (DANN), Combined Annotation-Dependent Depletion (CADD), Polymorphism Phenotyping v2 (PolyPhen-2), and Sorting Intolerant from Tolerant (SIFT). Variants were classified according to the American College of Human Society (ACMG) guidelines. Segregation was verified by Sanger sequencing in the families. Accession numbers are the following: human *ITPR1* mRNA: NM_001168272.1; human ITPR1 protein: NP_001161744.1.

## 3. Results

### 3.1. Demographic Data

Group A included 14 patients (8 females and 6 males) from 10 unrelated families, with average age at the last follow-up of 18 years (min 2 years; max 56 years). One patient received a diagnosis of SCA15 (patient 4), eleven had SCA29, while two (patient 6 and 10) had Gillespie syndrome.

Six patients (5 females and 1 male, all sporadic) with a superior cerebellar atrophy were initially identified but due to lack of DNA from one patient, Group B finally included five patients. The average age at last follow-up was 7 years (min 6 months; max 14 years).

All patients belonging to Group B had a SCA29 phenotype (see Table 1).

### 3.2. Genetic Data

Pathogenic or likely pathogenic variants in the *ITPR1* gene were overall detected in 16 patients, while 2 siblings carried a novel variant of unknown significance (VUS) (p.S695N; CADD 20.7), which was absent from our in-house database of over 2000 WES, as well as from the gnomAD population database (Figure 1, Table 1).

In Group A, only the p.S695N missense variant was novel, while all other missense variants (p.R269W, p.T267M, p.R241K, p.N2576I, p.A280D, p.E497K) as well as a one-amino acid deletion (p.K2596del) had already been reported [1,9,10,11].

In Group B, 5 patients were genetically tested, of whom 4 were found to carry the following *ITPR1* previously reported missense variants: p.R269W; p.T267M (2 unrelated patients); p.E497K [1,10].

### 3.3. Neuroradiological Data

In Group A, 11/14 patients showed a pattern of predominant superior cerebellar atrophy (very mild to severe) while 3/14 patients showed diffuse atrophy.

In cases of superior atrophy, both the upper part of the vermis and hemispheres were affected (Figure 2). Follow-up studies were available in three cases: in one case, diffuse cerebellar atrophy became evident between 5 months and 6 years of age (Figure 3A); in the second case, superior atrophy was not present at 8 months of age (Figure 3B) and it became evident at 3 years of age; in the last one, very mild superior cerebellar atrophy remained stable over a 4 year period (Figure 3D). Regarding signal alterations, a very mild hyperintensity of the superior cerebellar cortex close to the vermis was noticed in three cases on FLAIR but not on T2-weighted images. The significance of this finding remains unclear as it could be partially related to artifacts at the interface between cortex and enlarged CSF spaces. Supratentorial findings were mostly normal; in two patients from the same family, a dysmorphic corpus callosum with enlarged lateral ventricles was observed.

Patients of Group B were retrospectively selected according to MRI reports and images. They showed in 4/5 cases a clear pattern of superior cerebellar atrophy with almost normal inferior cerebellum, and in 1/5 a diffuse atrophy, more severe in the upper cerebellum (Figure 4). One patient had a follow-up scan that did not document any progression over a 2-year period between 12 and 14 years of age (Figure 3C).

Supratentorial findings were unremarkable in all cases.

Neuroradiological, genetic, and clinical data of all patients with *ITPR1* variants reported in the literature are summarized in Table 2.

## 4. Discussion

Imaging findings in *ITPR1*-mutated patients are sparsely reported in the literature, and not systematically addressed [8]. In the available studies, the most frequently described neuroradiological feature is a diffuse cerebellar atrophy, sometimes reported as predominantly involving the vermis, and just in a minority of cases involving the superior vermis and hemispheres [3,9,10,11,15].

In our study population, all three *ITPR1*-related phenotypes were represented. We included both patients previously diagnosed with a pathogenic or likely pathogenic *ITPR1* variant as well as patients who were directed to *ITPR1* genetic testing due to the presence of isolated superior vermian and hemispheric cerebellar atrophy. We also included two affected siblings carrying a novel *ITPR1* variant classified as VUS (p.S695N), since the clinical and neuroradiological phenotype in both patients were highly suggestive of an *ITPR1*-related defect. Unfortunately, the family is from Morocco and parents were not available for clinical examination and segregation analysis.

Considering all *ITPR1* mutated patients, a characteristic pattern of superior vermian and cerebellar atrophy was present in 83%, while in the remaining cases (3/18, 17%) a less peculiar diffuse cerebellar atrophy was noted. We searched for specific clinical features or a different severity manifestation in patients with diffuse cerebellar atrophy, but we could not identify differences compared to patients who showed the more typical superior cerebellar involvement [14,16,17,18,19].

*ITPR1* gene encodes for inositol 1,4,5-triphosphate receptor type 1, an InsP_3_-gated calcium channel that modulates intracellular Ca^2+^ release and plays the crucial role in the regulation of spine distribution and morphology of adult Purkinje cells [20]. The primary structure of the protein ITPR1 consists of three domains, including an InsP_3_-binding domain in the N-terminus, a regulatory carbonic anhydrase-related protein VIII (CARP)-binding domain, and a transmembrane-spanning domain near the C-terminus [1]. According to the literature, most of the pathogenic variants in our cohort (as the majority of reported variants) fall within the IRBIT binding domain, where lie most of the mutations associated with SCA29, while variants that cause Gillespie syndrome are mostly located at the C-terminus of the protein, especially in the transmembrane domain [9]. In our cohort, we observed intrafamiliar variability both in terms of clinical phenotype and neuroradiological pattern. For instance, in family V, the proband (patient 8) showed a phenotype compatible with SCA29, while the mother (patient 9) had a later-onset, milder phenotype with only slurred speech, suggesting a SCA15 phenotype. This evidence highlights the possible occurrence of distinct phenotypes (SCA15 and SCA29) in association with the same mutation, as previously suggested [1].

Of note, we observed a variable expression of the neuroradiological pattern among patients carrying the same *ITPR1* variant, with the only exception of carriers of the recurrent p.R269W variant, who showed a fully concordant imaging phenotype of superior cerebellar atrophy. Overall, we failed to detect reliable correlates between the protein domain harboring the mutation and the pattern of cerebellar atrophy observed (diffused vs. predominantly superior). We speculate that other regulatory factors might influence the pattern of expression of the protein in the cerebellum, as already suggested by Kerkhofs et al. [21].

Aside from the characterization of neuroimaging pattern, the analysis of Group B patients highlights the importance of recognizing superior cerebellar atrophy as a diagnostic clue for *ITPR1*-related disorders, since four out of five subjects presenting this peculiar imaging trait, retrospectively selected from two large imaging databases, tested positive for pathogenic variants in the *ITPR1* gene. Predominant atrophy of the upper parts of the cerebellum has never been previously associated to proven genetic conditions [22,23], while superior vermian atrophy has been described in neonates suffering from hypoxic-ischemia [24]. This represents a novel and key element for improving the diagnosis of children with either static or progressive cerebellar atrophy.

Clinical features of pediatric ataxia are usually unspecific and can be ascribed to a very large number of genetic conditions. Generally, the initial diagnostic workup, beyond a comprehensive clinical assessment of ataxic and non ataxic symptoms, includes neuroimaging and genetic investigations [25]. Therefore, considering our finding, an MRI pattern of mild to severe atrophy involving the superior part of cerebellar hemispheres and vermis (typically without any signal alterations within the cortex), with normal supratentorial brain and without history and cerebral signs of hypoxic-ischemic injury, might represent a very important insight and should prompt suspicion of a possible *ITPR1* gene-related disorder and genetic testing is highly recommended.

The retrospective nature of this study represents a limitation for an even more extensive definition of the imaging spectrum of *ITPR1*-mutated patients. For instance, due to the lack of seriate MRIs in all patients, we could not establish the presence and severity of atrophy at symptoms’ onset, nor we could assess its progression over time. The few cases with follow-up imaging suggest that atrophy is not evident in the very first months of life, but it becomes evident during early childhood and may remain stable afterward. However, further confirmation is needed for this preliminary observation.

In conclusion, through a careful review of MRI images, we demonstrated a peculiar pattern of cerebellar atrophy in patients with *ITPR1* gene defects and we propose that it is highly suggestive for the diagnosis, which might orient the choice of genetic testing.

## Figures and Tables

**Figure 1 ijms-23-06723-f001:**
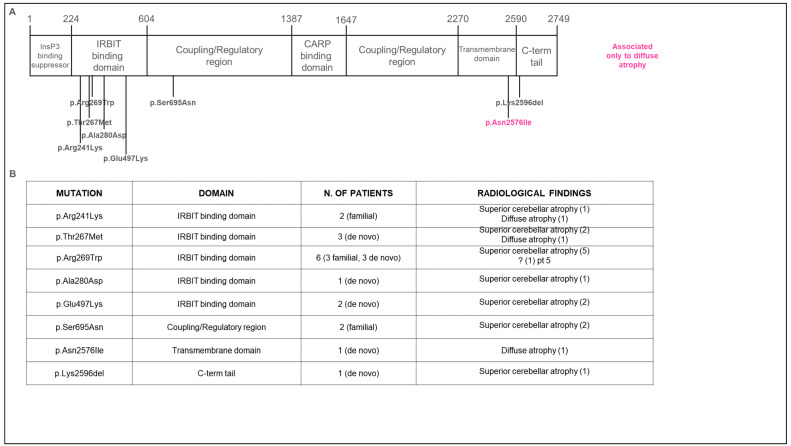
Genetic findings in enrolled patients. (**A**) Schematic representation of the position of variants within functional domains of the ITPR1 protein. Most variants are located in the IRBIT domain, suggesting that loss of the channel function impairs the IP3-induced Ca^2+^ release. (**B**) Summary of variants and radiological features found in our cohort.

**Figure 2 ijms-23-06723-f002:**
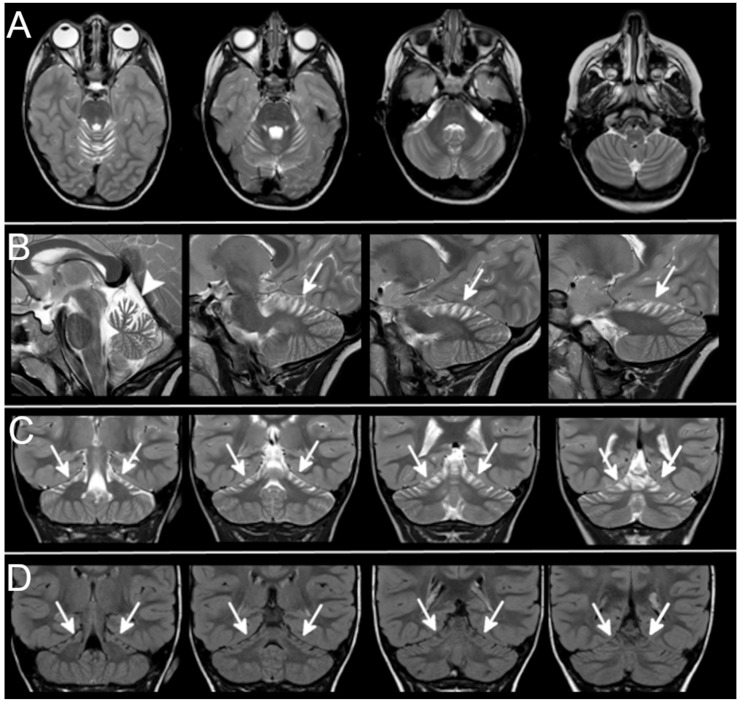
Superior cerebellar atrophy. Axial (**A**), sagittal (**B**), coronal (**C**), T2-weighted and coronal FLAIR (**D**) sections show the typical pattern of superior cerebellar atrophy (Patient 18-Table 1). The superior part of cerebellar hemispheres (arrows) and vermis (arrowhead) show marked atrophy with enlarged cortical CSF spaces. No cerebellar signal alterations can be detected on T2-weighted and FLAIR sections. The inferior part of the cerebellum is not atrophic and looks normal.

**Figure 3 ijms-23-06723-f003:**
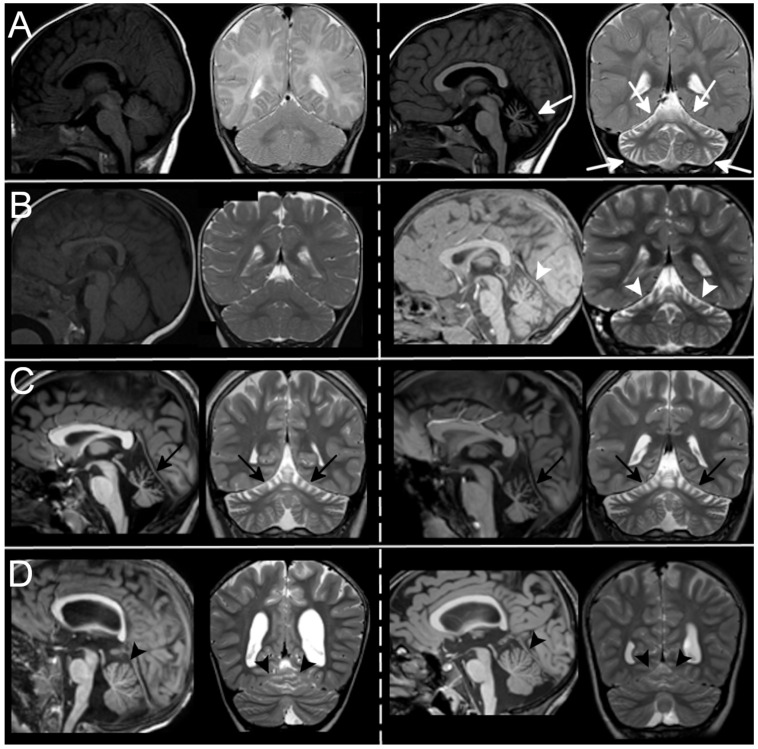
MRI evolution over time. In (**A**), Patient7-Table 1 has a normal cerebellum at 5 months of age (left) while he shows a diffuse cerebellar atrophy (white arrows) at 6 years of age (atrophy was also present at 2 years of age. Not shown here). In (**B**), Patient 12-Table 1 develops superior cerebellar atrophy (white arrowheads) between 8 months of age (left, normal cerebellum), and 3 years (right, cerebellar atrophy). No progression of atrophy is seen in Patients 1 and 18-Table 1, with moderate ((**C**), black arrows) and very mild ((**D**), black arrowheads) superior cerebellar atrophy within a 2-year and 4-year period respectively.

**Figure 4 ijms-23-06723-f004:**
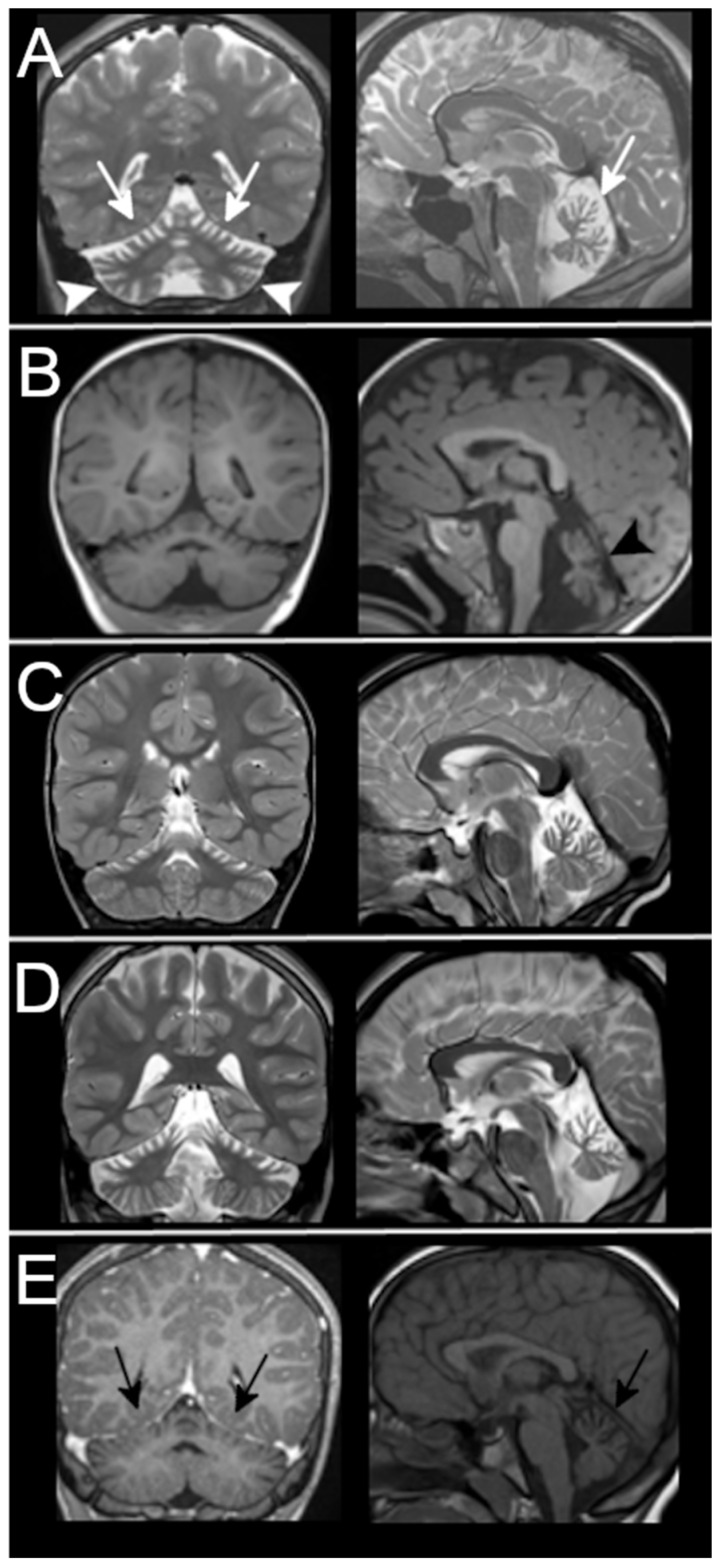
MRI findings in patients from Group B. MRIs of the 5 patients retrospectively selected according to the imaging pattern are shown here. All of them have superior cerebellar atrophy. Patient 14-Table 1 in (**A**) shows diffuse cerebellar atrophy that also involves the inferior cerebellum (arrowheads) but that is more severe in the upper part of vermis and hemispheres (white arrows). The patient with mild superior atrophy (black arrows) in (**E**) tested negative for *ITPR1* gene defects.

**Table 1 ijms-23-06723-t001:** Clinical, neuroradiological and genetic features of enrolled patients.

Group A									
Family	Patient	Gender; Age at Last Evaluation	Clinical Features Phenotype	Superior Cerebellar Atrophy	Diffuse Atrophy	Progression over Time	Genetics	ACMG Classification	CADD; DANN
I	1. Proband	M; 9 years	DD, ID, hypotonia, ataxia, facial dysmorphisms, cryptorchidism; SCA29	+	-	No	c2084G > A; p.S695N	Uncertain significance	20.7; 0.9919
	2. Sister	F; 7 years	DD, ID, hypotonia, ataxia; SCA29	+	-	NA			
II	3. Proband *	M; 28 years	Severe motor delay, normal cognitive level, hypotonia, ataxia, postural tremor, slurred speech, nystagmus, OMA; SCA29	+	-	NA	c.805C > T; p.R269W	Pathogenic (ClinVar: Pathogenic) [1]	26.399; 0.9992
	4. Mother *	F; 56 years	Motor delay, normal cognitive level, hypotonia, ataxia, slurred speech, OMA; SCA15	+	-	NA			
	5. Brother *	M; 23 years	Severe motor delay, mild ID, hypotonia, ataxia, slurred speech, nystagmus; SCA29	+	-	NA			
III	6. Proband	F; 2 years	Ambulation not achieved, mild ID, hypotonia, ataxia, nystagmus, bilateral iris hypoplasia; Gillespie syndrome	+	-	NA	c.7786-7788delAAG p.K2596del de novo	Pathogenic [9]	//
IV	7. Proband	M; 13 years	DD, mild ID, hypotonia, ataxia, tremor, nystagmus, OMA; SCA29	-	+	Yes	c.800C > T p.T267M de novo	Pathogenic (ClinVar: Pathogenic) [10]	26.2; 0.9993
V	8. Proband *	F; 19 years	Severe motor delay, normal cognitive level, hypotonia, ataxia, postural tremor, slurred speech, nystagmus, OMA; SCA29	-	+	NA	c.722G > A p.R241K	Pathogenic (ClinVar: Pathogenic) [1]	28.1; 0.9954
	9. Mother *	F; 42 years	Slurred speech, normal cognitive level; SCA29	+	-	NA			
VI	10. Proband *	F; 29 years	Severe motor delay, moderate ID, hypotonia, ataxia, slurred speech, nystagmus, bilateral iris hypoplasia, ptosis; Gillespie syndrome	-	+	NA	c.7727A > T p.N2576I de novo	Likely pathogenic [11]	28.5; 0.9913
VII	11. Proband	M; 12 years	Hypotonia, ataxia, dysarthria, nystagmus, OMA; SCA29	+	-	NA	c.805C > T; p.R269W de novo	Pathogenic (ClinVar: Pathogenic) [1]	26.399; 0.9992
VIII	12. Proband	F; 3 years	DD, hypotonia, ataxia, nystagmus, dysarthria, OMA; SCA29	+	-	Yes	c.805C > T; p.R269W de novo	Pathogenic (ClinVar: Pathogenic) [1]	26.399; 0.9992
IX	13. Proband *	M; 6 years	Moderate motor delay, normal cognitive level, hypotonia, ataxia, postural tremor, slurred speech, nystagmus; SCA29	+	-	NA	c.839C > A p.A280D de novo	Pathogenic [1]	28.2; 0.9976
X	14. Proband *	F; 7 years	Severe motor delay, normal cognitive level, hypotonia, ataxia, postural tremor, slurred speech, nystagmus; SCA29	+	-	NA	c.1488G > A p.E497K de novo	Likely pathogenic (ClinVar: Likely Pathogenic) [1]	29.2; 0.9994
**Group B**									
XI	15. Proband	F; 6 years	Severe motor delay, ataxia, hypotonia, nystagmus SCA29	+	-	NA	c.805C > T p.R269W de novo	Pathogenic (ClinVar: Pathogenic) [1]	26.399; 0.9992
XII	16. Proband	F; 18 months	Severe motor delay, hypotonia, ataxia, OMA; SCA29	+	-	NA	c.800C > T p.T267M de novo	Pathogenic (ClinVar: Pathogenic) [10]	26.2; 0.9993
XIII	17. Proband	F, 14 years	progressive spastic paraparesis; SCA29	+	-	NA	negative	//	//
XIV	18. Proband	F, 7 years	Moderate motor delay, moderate ID, hypotonia, ataxia, slurred speech, nystagmus; SCA29	+	-	NA	c.1489G > A p.E497K de novo	Likely pathogenic (ClinVar: Likely Pathogenic) [1]	29.2; 0.9994
XV	19. Proband	M, 12 years	Ambulation not achieved, moderate ID hypotonia, ataxia, slurred speech, nystagmus, OMA; SCA29	+	-	No	c.800C > T p.T267M de novo	Pathogenic (ClinVar: Pathogenic) [10]	26.2; 0.9993

* Previously published patients [1,11]; +: present; -: absent; NA: not available; DD: developmental delay; ID: intellectual disability; OMA: ocular-motor apraxia; SCA: spinocerebellar ataxia.

**Table 2 ijms-23-06723-t002:** Gene mutations described up to date.

Study/Journal	Number of Patients	Age Range	Phenotype	Infratentorial Imaging	Associated Neuroradiological Findings	Progression of Cerebellar Atrophy
Hara et al., 2008 [10] *Neurology*	2 families (10 affected members)	NA	SCA15	Cerebellar atrophy	-	NA
Di Gregorio et al., 2010 [11] *Cerebellum*	2 families (12 affected members)	44–81 years	SCA15 (buccolingual dyskinesias, facial myokymias, pyramidal signs)	Cerebellar vermis atrophy with a mild involvement of the hemispheres in some individuals	-	NA
Novak et al., 2010 [5] *Mov. Disord.*	1 family (3 affected members)	38–56 years	SCA15	Moderate cerebellar atrophy, which preferentially involves the superior vermis	Cortical parietal and temporal atrophy	NA
Huang et al., 2012 [2] *Orphanet J. Rare Dis.*	1 family (3 affected members)	5–45 years	SCA29	Mild cerebellar vermis atrophy	-	Yes
Sasaki et al., 2015 [4] *J. Neurol.*	4 patients	6–12 years	SCA15, SCA29	Superior cerebellar hemispheres atrophy, vermian diffuse atrophy	Atrophy of the pontine tegmentum	Yes
Gerber et al., 2016 [7] *Am. J. Hum. Genet.*	5 patients	1.5–18 years	Gillespie syndrome	Cerebellar atrophy	Thin CC	Yes
Mc Entagart et al., 2016 [9] *Am. J. Hum. Genet.*	13 patients	13–55 years	Gillespie syndrome	Atrophy mainly affecting the superior vermis, involving superior cerebellar hemispheres more than the inferior	Abnormal periventricular increased T2/FLAIR white matter signal adjacent to the frontal horns	Yes
Shadrina et al., 2016 [12] *Cerebellum Ataxias*	1 family (2 affected members)	54 years	SCA15	Mild cerebellar atrophy	-	NA
Barresi et al., 2017 [1] *Clin. Genet.*	4 families (6 affected members)	7–28 years	SCA29, SCA15	Cerebellar and/or vermis atrophy	-	Yes
Dentici et al., 2017 [13] *Gene*	2 patients	2–29 years	Gillespie syndrome	Cerebellar atrophy, predominantly in the vermis	-	NA
Klar et al., 2017 [3] *Eur. J. Hum. Genet.*	Family (6 affected members)	17–45 years	SCA29	Cerebellar atrophy most pronounced in the vermis	-	NA
Van Dijk et al., 2017 [8] *Am. J. Med. Genet.*	1 patient	6 years	SCA29	The vermis inferior is almost absent and the vermis superior showed hypoplasia with superimposed atrophy. The vermis inferior is almost absent and the vermis superior showed hypoplasia with superimposed atrophy. Almost absent inferior vermis, hypoplasia with atrophy of the superior vermis	Hyperintensities in medulla oblungata	No
Zambonin et al., 2017 [6] *Orphanet J. Rare Dis.*	21 patients	28 m–49 years	SCA29	Cerebellar atrophy, often with superior cerebellar hemispheres and vermis	Pontine atrophy	Yes
Paganini et al., 2018 [14] *Am. J. Med. Genet.*	1 family (2 affected members)	6–9 years	Gillespie syndrome	Generalized atrophy, mainly vermis atrophy	-	Yes
Synofzik et al., 2018 [15] *Eur. J. Hum. Genet.*	5 families (10 affected members)	33–80 years	SCA15	Cerebellar atrophy with a major involvement of vermis	-	NA
Wang et al., 2018 [16] *Cerebellum*	4 patients	6–51 years	SCA29	Cerebellar hemisphere atrophy	-	NA
Stendel et al., 2019 [17] *Neuropediatrics*	1 patient	NA	Gillespie syndrome	Atrophy of the anterior cerebellar vermis	-	NA

CC: corpus callosum; NA: not available; SCA: spinocerebellar ataxia.

## Data Availability

Not applicable.

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
