# Peer review of "Superior Cerebellar Atrophy: An Imaging Clue to Diagnose ITPR1-Related Disorders"

_ijms, 2022, doi:10.3390/ijms23126723_

Round 1
Reviewer 1 Report
The paper by Romaniello and colleagues presents and interesting retrospective study on the potential effect of ITPR1 gene in modulating cerebellar functions. In particular, it has been suggested a relationship to various types of spinocerebellar ataxias. Specifically, this study is focused on the relationship between ITPR1 gene expression and MRI findings, thus suggesting to use neuroradiological patterns for hypothesizing gene pathogenic variants. Although limited by its retrospective nature, the study is sound and relevant in clinical settings.
I would have appreciated a more extended discussion on the functional and behavioural characteristics of spinocerebellatr ataxias associated to ITPR1gene. These are present in the introduction but largely absent in the discussion. Although the paper is focused on genetic and neuroimaging data, a stronger relationship with behavioural patterns may be relevant in clinical settings.
Author Response
Reviewer 1
The paper by Romaniello and colleagues presents and interesting retrospective study on the potential effect of ITPR1 gene in modulating cerebellar functions. In particular, it has been suggested a relationship to various types of spinocerebellar ataxias. Specifically, this study is focused on the relationship between ITPR1 gene expression and MRI findings, thus suggesting to use neuroradiological patterns for hypothesizing gene pathogenic variants. Although limited by its retrospective nature, the study is sound and relevant in clinical settings. I would have appreciated a more extended discussion on the functional and behavioural characteristics of spinocerebellar ataxias associated to ITPR1gene. These are present in the introduction but largely absent in the discussion. Although the paper is focused on genetic and neuroimaging data, a stronger relationship with behavioural patterns may be relevant in clinical settings.
Reply: the Authors thank the Reviewer for the positive comment to our manuscript.
With regard to the request “to extend discussion…ITPR1 gene … a stronger relationship with behavioral patterns may be relevant in clinical setting”, the Authors point out that the functional and behavioral pattern associated with ITPR1 gene related disorders to date described in literature is quite aspecific and not so relevant in clinical setting to relate to genetic and neuroimaging data in the discussion. Moreover, these aspects go beyond the aim of the study that the Authors claimed to be “to describe ITPR1-related neuroimaging findings”.
Reviewer 2 Report
Romaniello and colleagues in this research paper address a highly relevant topic in neuropediatrics, which is the possibility of identifying cerebellar atrophy as a predictive cue for the early diagnosis of neurological disorders associated to ITPR1 gene mutations. This gene, coding for a ligand-gated calcium channel and highly expressed in cerebellar Purkinje cells, has been associated to distinct cerebellar diseases, such as SCA29, SCA15 and Gillespie syndrome.
After evaluating, through the analysis of fMRI scans, the incidence and severity of cerebellar atrophy in a cohort of patients previously diagnosed with distinct ITPR1 mutations and after addressing whether any mutation was indeed present in other patients with recognized cerebellar atrophy, the authors conclude that atrophy is indeed a good predictive sign for diagnosis.
In principle, the topic addressed is very important in view of identifying at the disease onset the presence of genetic mutations and in view of an early intervention to minimize disease severity.
Nevertheless, several aspects need further clarification to fully support the authors’ conclusions and to make this paper suitable for publication.
1. The evaluation of the presence and entity of cerebellar atrophy is merely qualitative. The authors provide scan images with arrows pointing to atrophic regions but it is difficult for non-experts to evaluate the entity of the atrophy from these images. Moreover, since this is a key aspect in the paper, measurements should be provided, compared to age-matched control cerebella to evaluate in an unbiased way the degree of atrophy.
2. Along the same line, at line 183, the authors claim that “The patient with mild superior atrophy (black arrows) in E tested negative for ITPR1 gene defects”: this statement, in view of the general aim of the paper, appears really important and should be properly discussed in the text. Indeed, it may suggest that atrophy under a certain level is not predictive for ITPR1 mutations and associated diseases. Therefore, quantifying the differences between what is considered “mild” and what is considered a more severe atrophy could be highly relevant.
3. Similarly, measurements of the cerebella acquired in the follow ups (Figure 3) are missing and should be provided.
4. By looking at the diagnosis of the distinct patients belonging to both cohorts A and B, it turns out that most of them have a SCA29 phenotype. SCA29 patients are indeed 11 out of 18 in group A and, importantly, 4 out of 5 in group B. For this reason, the conclusion that atrophy is a good predictive clue for the diagnosis of “ITPR1gene-related disorders” appears overall an overstatement. Rather, from the available evidence the authors could more likely conclude that it can be a good hint to diagnose ITPR1-related SCA29. To make this conclusion suitable for other diseases associated to the same gene, the authors should increase the number of patients analyzed in group B, including also non-SCA29 phenotypes. Otherwise, the authors should consider of re-phrasing the conclusions and the main title.
5. From the available studies cited by the authors and from the cohorts of patients analyzed in this paper, we can assume that cerebellar atrophy is a hallmark of those diseases associated with ITPR1 mutations. Nevertheless, as unveiled by the follow up studies in lines 152-155, this clinical sign does not appear to be an early onset sign, although its exact onset could not be assessed. Therefore, one may speculate that other clinical signs, appearing before 8 months of age when cerebellar atrophy is still not present, may be better, earlier, predictors of ITPR1 mutations. For instance, hypotonia appears in nearly all the patients analyzed. Is this (or others) clinical symptom already evident before the appearance of cerebellar atrophy? The authors should better address this point, carefully comparing the onset of other clinical signs vs cerebellar atrophy, and better discuss why cerebellar atrophy should be the best predictor, although not present in the early phases of disease.
Other minor comments:
1. The aims of the study and the added value provided by its main findings should be more clearly stated in the introduction section. Why should it be important to identify an early cue for genetic mutations in the discussed pathologies? How could the diagnosed patients benefit from this added information under the therapeutic point of view?
2. To contextualize the genetic mutation in the frame of the disease, the authors should comment about the role played by this gene in the cerebellum, and especially in Purkinje cells development and functioning (https://www.jneurosci.org/content/33/30/12186).
3. In table 1 the information concerning the diagnosis of the patients as explained in lines 113-115 is missing and should be added to make the whole scenario clearer.
4. Line 119: “All patients had a SCA29 phenotype except patient 4”. There is no patient 4 in the group b (patients from 11 to 15 according to table 1), please re-number the patient in the text accordingly.
6. Figure 2 in the text is recalled in the frame of group A description but shows patient n.17, which belongs to the B cohort. This creates confusion and should be changed accordingly.
7. The second table was erroneously named Table 1.
Author Response
Reviewer 2
Romaniello and colleagues in this research paper address a highly relevant topic in neuropediatrics, which is the possibility of identifying cerebellar atrophy as a predictive cue for the early diagnosis of neurological disorders associated to ITPR1 gene mutations. This gene, coding for a ligand-gated calcium channel and highly expressed in cerebellar Purkinje cells, has been associated to distinct cerebellar diseases, such as SCA29, SCA15 and Gillespie syndrome. After evaluating, through the analysis of fMRI scans, the incidence and severity of cerebellar atrophy in a cohort of patients previously diagnosed with distinct ITPR1 mutations and after addressing whether any mutation was indeed present in other patients with recognized cerebellar atrophy, the authors conclude that atrophy is indeed a good predictive sign for diagnosis. In principle, the topic addressed is very important in view of identifying at the disease onset the presence of genetic mutations and in view of an early intervention to minimize disease severity.
Nevertheless, several aspects need further clarification to fully support the authors’ conclusions and to make this paper suitable for publication.
Reply: the Authors thank the Reviewer for the positive comment to our manuscript.
Below the point-by-point answers
- 1. The evaluation of the presence and entity of cerebellar atrophy is merely qualitative. The authors provide scan images with arrows pointing to atrophic regions but it is difficult for non-experts to evaluate the entity of the atrophy from these images. Moreover, since this is a key aspect in the paper, measurements should be provided, compared to age-matched control cerebella to evaluate in an unbiased way the degree of atrophy.
- Along the same line, at line 183, the authors claim that “The patient with mildsuperior atrophy (black arrows) in E tested negative for ITPR1 gene defects”: this statement, in view of the general aim of the paper, appears really important and should be properly discussed in the text. Indeed, it may suggest that atrophy under a certain level is not predictive for ITPR1 mutations and associated diseases. Therefore, quantifying the differences between what is considered “mild” and what is considered a more severe atrophy could be highly relevant.
- Similarly, measurements of the cerebella acquired in the follow ups (Figure 3) are missing and should be provided.
Reply to point 1-3: The Authors point out that cerebellar atrophy in literature has always been characterized qualitatively and not quantitatively (Poretti et al 2015 doi: 10.1055/s-0035-1564620. Epub 2015 Oct 7; Poretti and Boltshauser 2015 doi: 10.1186/s40673-015-0027-x); unfortunately full standardized measurement and referral normal values of posterior fossa structures in pediatric population are not available. The most relevant paper in the field was published by Jandeaux et al in 2019 (doi: 10.3174/ajnr.A6257. Epub 2019 Oct 17). Jandeaux et al. provide normal values for 2D measurements of vermis and pons on the midsagittal slice in children. In our cohort such measures would not pick up the atrophy of cerebellar hemispheres and the peculiar involvement of the superior part of the cerebellum. Moreover, due to the retrospective nature of our study, our patients underwent the neuroimaging exams in different Centers, and MRI sequences and exam quality show an interindividual variability. For these reasons a quantitative analysis, which could reinforce our results, is unfortunately not doable.
- 4. By looking at the diagnosis of the distinct patients belonging to both cohorts A and B, it turns out that most of them have a SCA29 phenotype. SCA29 patients are indeed 11 out of 18 in group A and, importantly, 4 out of 5 in group B. For this reason, the conclusion that atrophy is a good predictive clue for the diagnosis of “ITPR1gene-related disorders” appears overall an overstatement. Rather, from the available evidence the authors could more likely conclude that it can be a good hint to diagnose ITPR1-related SCA29. To make this conclusion suitable for other diseases associated to the same gene, the authors should increase the number of patients analyzed in group B, including also non-SCA29 phenotypes. Otherwise, the authors should consider of re-phrasing the conclusions and the main title.
Reply: The Authors thanks the Reviewer for the comment. Moreover they point out that the term “clue” means an “indication” and not an hallmark for diagnosis so they consider appropriate to modify the statement in conclusion as follows:” the superior cerebellar atrophy is highly suggestive for the diagnosis….”as requested, but not the main title.
- From the available studies cited by the authors and from the cohorts of patients analyzed in this paper, we can assume that cerebellar atrophy is a hallmark of those diseases associated with ITPR1 mutations. Nevertheless, as unveiled by the follow up studies in lines 152-155, this clinical sign does not appear to be an early onset sign, although its exact onset could not be assessed. Therefore, one may speculate that other clinical signs, appearing before 8 months of age when cerebellar atrophy is still not present, may be better, earlier, predictors of ITPR1 mutations. For instance, hypotonia appears in nearly all the patients analyzed. Is this (or others) clinical symptom already evident before the appearance of cerebellar atrophy? The authors should better address this point, carefully comparing the onset of other clinical signs vs cerebellar atrophy, and better discuss why cerebellar atrophy should be the best predictor, although not present in the early phases of disease.
Reply: In this manuscript the Authors suggest to consider superior cerebellar atrophy as a possible specific pattern of cerebellar atrophy associated to ITPR1 gene mutation. The common neurological signs and clinical data recurring in the study population include hypotonia, ataxia, motor delay, ataxic speech, nystagmus. None of these signs can be considered ‘specific’ for ITPR1 diagnosis. What can orient in the clinical and genetic diagnosis in SCA is indeed a multi-componential assessment of disease onset/presentation, cerebellar and extracerebellar clinical signs, neuroimaging, neurophysiological data, and laboratory data (Brandsma et al 2019, https://doi.org/10.1016/j.ejpn.2019.08.004).
Other minor comments:
- The aims of the study and the added value provided by its main findings should be more clearly stated in the introduction section. Why should it be important to identify an early cue for genetic mutations in the discussed pathologies? How could the diagnosed patients benefit from this added information under the therapeutic point of view?
Reply: the Authors thank the Reviewer for the comment. The importance to identify an early cue is well known in current medicine, as it has fundamental implications for genetic counseling and targeted rehabilitation aspects, despite to date in this case specific therapeutic strategies do not exist yet.
- To contextualize the genetic mutation in the frame of the disease, the authors should comment about the role played by this gene in the cerebellum, and especially in Purkinje cells development and functioning (https://www.jneurosci.org/content/33/30/12186).
Reply: the Authors thank the Reviewer for the comment and have added in the discussion the follow sentence “…and plays the crucial role in the regulation of spine distribution and morphology of adult Purkinje cells [20]” and the indicated reference.
- In table 1 the information concerning the diagnosis of the patients as explained in lines 113-115 is missing and should be added to make the whole scenario clearer.
Reply: the Authors thank the Reviewer for the comment. They have clarified in the text that all patients have a diagnosis of SCA29 except patient 4 who show a SCA15 phenotype and patient 6 and 10 who have a diagnosis of Gillespie syndrome. These data have been added in table I as requested
- Line 119: “All patients had a SCA29 phenotype except patient 4”. There is no patient 4 in the group b (patients from 11 to 15 according to table 1), please re-number the patient in the text accordingly.
Reply: the Authors thank the Reviewer for the comment, The text has been edited and clarified it as requested: All patients belonging to Group b had a SCA29 phenotype (see Table I)”.
- Figure 2 in the text is recalled in the frame of group A description but shows patient n.17, which belongs to the B cohort. This creates confusion and should be changed accordingly.
Reply: the Authors thank the Reviewer for the comment. The text has been edited “In Group a, 11/14 patients showed a pattern of predominant superior cerebellar atrophy (very mild to severe) while 3/14 patients showed diffuse atrophy. In cases of superior atrophy, both the upper part of the vermis and hemispheres were affected (Figure 2).” and Legend Figure 2 is edited as requested (Patient 18 instead of Patient 17)
- The second table was erroneously named Table 1.
Reply: the Authors thank the Reviewer for the comment. The second table was namely correctly Table II.
Round 2
Reviewer 2 Report
In the previous review report, this reviewer highlighted that, for an unbiased evaluation of the degree of cerebellar atrophy within the sample analyzed, quantitative measurements would have been needed. This was the major concern raised that prevented the paper for publication in the submitted form. Nevertheless, the authors have now clarified that standardized measurements for cerebellar atrophy currently do not exist and could not in any case be applied to this retrospective study.
Moreover, the authors addressed the other raised points in a quite satisfactory way. Still, as a follow up of point 5, I would suggest to underline that most, if not all, the signs showed by the analyzed patients (i.e. ataxia, hypotonia, motor delay etc) are a common feature with other diseases in which cerebellar atrophy is not present (or is present, but with lower incidence), citing the relevant literature. This would clarify that this sign is indeed a very good cue, likely specific for ITPR1-related diseases. In the present form, this concept is still abstract and not clearly addressed in any section of the text.
Provided the relevance of the message conveyed by this paper, which is the possibility to find a cue for the early diagnosis of SCA29 and possibly other ITPR1-associated diseases, this reviewer is now convinced that the paper can be accepted for publication with only the minor adjustment suggested above.
Author Response
Reviewer 2
In the previous review report, this reviewer highlighted that, for an unbiased evaluation of the degree of cerebellar atrophy within the sample analyzed, quantitative measurements would have been needed. This was the major concern raised that prevented the paper for publication in the submitted form. Nevertheless, the authors have now clarified that standardized measurements for cerebellar atrophy currently do not exist and could not in any case be applied to this retrospective study. Moreover, the authors addressed the other raised points in a quite satisfactory way. Still, as a follow up of point 5, I would suggest to underline that most, if not all, the signs showed by the analyzed patients (i.e. ataxia, hypotonia, motor delay etc) are a common feature with other diseases in which cerebellar atrophy is not present (or is present, but with lower incidence), citing the relevant literature. This would clarify that this sign is indeed a very good cue, likely specific for ITPR1-related diseases. In the present form, this concept is still abstract and not clearly addressed in any section of the text. Provided the relevance of the message conveyed by this paper, which is the possibility to find a cue for the early diagnosis of SCA29 and possibly other ITPR1-associated diseases, this reviewer is now convinced that the paper can be accepted for publication with only the minor adjustment suggested above.
Reply: The authors thank the reviewer for the comment. The text has been modified accordingly: “Clinical features of pediatric ataxia are usually unspecific and can be ascribed to a very large number of genetic conditions. Generally, the initial diagnostic workup, beyond a comprehensive clinical assessment of ataxic and non ataxic symptoms, includes neuroimaging and genetic investigations [25]. Therefore, considering our finding, an MRI pattern of mild to severe atrophy involving the superior part of cerebellar hemispheres and vermis (typically without any signal alterations within the cortex), with normal supratentorial brain and without history and cerebral signs of hypoxic-ischemic injury, might represent a very important insight and should prompt to suspect a possible ITPR1 gene-related disorder and genetic testing is highly recommended” and a reference was added (25: Brandsma et al. Eur J Paediatr Neurol 2019;23(5):692-706. doi: 10.1016/j.ejpn.2019.08.004).